# Employment Status and Alcohol-Attributable Mortality Risk—A Systematic Review and Meta-Analysis

**DOI:** 10.3390/ijerph19127354

**Published:** 2022-06-15

**Authors:** Celine Saul, Shannon Lange, Charlotte Probst

**Affiliations:** 1Heidelberg Institute of Global Health (HIGH), Medical Faculty and University Hospital, Heidelberg University, 69120 Heidelberg, Germany; celine.saul@uni-heidelberg.de; 2Institute for Mental Health Policy Research, Centre for Addiction and Mental Health (CAMH), 33 Ursula Franklin Street, Toronto, ON M5S 2S1, Canada; shannon.lange@camh.ca; 3Department of Psychiatry, University of Toronto, Toronto, ON M5T 1R8, Canada; 4Campbell Family Mental Health Research Institute, Centre for Addiction and Mental Health (CAMH), Toronto, ON M5T 1R8, Canada

**Keywords:** employment status, socioeconomic status, inequality, alcohol-attributable mortality, systematic literature review and meta-analysis, alcohol use, mortality, public health

## Abstract

Being unemployed has been linked to various health burdens. In particular, there appears to be an association between unemployment and alcohol-attributable deaths. However, risk estimates presented in a previous review were based on only two studies. Thus, we estimated updated sex-stratified alcohol-attributable mortality risks for unemployed compared with employed individuals. A systematic literature search was conducted in August 2020 using the following databases: Embase, MEDLINE, PsycINFO, and Web of Science. The relative risk (RR) of dying from an alcohol-attributable cause of death for unemployed compared with employed individuals was summarized using sex-stratified random-effects DerSimonian-Laird meta-analyses. A total of 10 studies were identified, comprising about 14.4 million women and 19.0 million men, among whom there were about 3147 and 17,815 alcohol-attributable deaths, respectively. The pooled RRs were 3.64 (95% confidence interval (CI): 2.04–6.66) and 4.93 (95% CI 3.45–7.05) for women and men, respectively. The findings of our quantitative synthesis provide evidence that being unemployed is associated with an over three-fold higher risk of alcohol-attributable mortality compared with being employed. Consequently, a global public health strategy connecting brief interventions and specialized care with social services assisting those currently unemployed is needed.

## 1. Introduction

Social determinants of health, that is “the conditions in which people are born, grow, live, work and age” [1], are of increasing importance in health care and prevention. Being at the lower end of the socioeconomic spectrum, traditionally indicated by low income, education, or occupational status, is associated with poorer health compared with individuals at the upper end of the spectrum [2]. To address the root causes of highly unequal health outcomes, the World Health Organization established the Commission on Social Determinants of Health in 2005 [3]. Given the persistence of such inequalities, their reduction was also indicated as a global goal in the United Nations Sustainable Development Goals [4].

Although there are a number of core social determinants of health, employment status has been identified as especially important with respect to health outcomes [5]. Unemployment has been linked to detrimental health effects and heightened mortality risks. Specifically, being unemployed is associated with poorer physical health [5] and adverse mental health outcomes [6]. Further, unemployment has repeatedly been shown to be associated with increased all-cause mortality risk [7,8,9,10,11,12,13]. For instance, the hazard of all-cause mortality for those unemployed was increased by an amount equivalent to 10 extra years of age, compared with those employed [14]. Moreover, Roelfs and colleagues [12] found that unemployment was associated with a 63% increase in all-cause mortality risk.

It was forecasted that following a period of decline, the global unemployment rate in 2019 (5.4%) would remain stable for at least two years [15]. Then, unexpectedly, the coronavirus disease 2019 (COVID-19) pandemic resulted in a massive global unemployment crisis, causing the rate to increase to 6.3% in 2021 [16]. According to the International Labour Organization, approximately 33 million people became unemployed due to the COVID-19 crisis. Thus, the negative health outcomes associated with being unemployed are likely to increase throughout the world.

A simulation study [8] predicted that the economic recession due to the COVID-19 pandemic would result in 0.84 million additional deaths over the next 15 years. Although there is a notable impact of economic downturns and rising unemployment rates on all-cause mortality, their impact on alcohol-attributable mortality may be even higher. A review conducted in 2013 reported a 1.5- to 2-fold higher mortality risk for alcohol-attributable causes compared with all causes for individuals with lower socioeconomic status (SES) [17]. Another review examining alcohol-attributable mortality risks showed that unemployment in particular is associated with relative risks (RR) of 6.1 and 12.3 for women and men, respectively [18]. While there appears to be a link between unemployment and alcohol-attributable mortality risks, this review was based on only two studies [19,20]. Therefore, the present work aims to provide updated, sex-stratified estimates on the relative alcohol-attributable mortality risk depending on employment status based on the current evidence.

## 2. Methods

The current study presents a subset of data from a larger systematic review and meta-analysis; data on other socioeconomic variables are reported elsewhere [21]. In this first review, we performed dose–response meta-analyses for all indicators of SES except employment status, as that is most often dichotomous (e.g., unemployed vs. employed) and does not lend itself well to a dose-response investigation. The study protocol of the present systematic review and meta-analysis followed the Preferred Reporting Items for Systematic Reviews and Meta-Analyses (PRISMA [22], Appendix A) and was preregistered in PROSPERO (registration number CRD42019140279).

### 2.1. Systematic Literature Search

Embase, MEDLINE, PsycINFO, and Web of Science were searched from February 2013 until the last week of August 2020, updating a previous systematic review that included all studies published up to February 2013 [18]. The studies identified in the previous review were reconsidered for inclusion. Studies reporting on alcohol-attributable mortality among unemployed compared with employed individuals (or individuals having the highest level of occupation) of the general adult population were included. Search terms relating to alcohol consumption, mortality, employment status, and study design were used and adapted to each of the databases searched (see Text S1). We manually screened reference lists and cited articles of all identified studies. We did not apply any geographical or language restrictions.

### 2.2. Study Selection and Inclusion Criteria

Studies were eligible for inclusion if they consisted of original, quantitative research reporting on the relative alcohol-attributable mortality risk by employment status, including a measure of uncertainty (confidence interval [CI] or standard error) or sufficient original data to calculate the risk and/or uncertainty. Alcohol-attributable causes of death were defined as all underlying causes of death that are fully attributable to alcohol use [23]. Studies that included causes with an alcohol-attributable fraction (AAF) of at least 10% globally [24] (see Appendix A, and Appendix A for ICD-codes) in addition to 100% attributable causes were also eligible. Study samples had to be based on the general adult (at least 15 years of age) population. Studies that used a longitudinal design with data-linkage, a cross-sectional design (deaths with a population denominator), or case-control design were eligible. For detailed inclusion and exclusion criteria, see Appendix A.

Titles and abstracts were screened by three reviewers to exclude records with high certainty. Next, full texts of all potentially eligible records were assessed for inclusion. Reviewer consensus meetings were held to discuss inclusion in cases where eligibility was unclear or reviewers disagreed. To avoid double counting of individuals, studies reporting on overlapping or identical data were excluded, giving preference to age-adjusted and sex-stratified estimates.

### 2.3. Data Extraction

We extracted data on study population, study design, mortality assessment, employment status, sample size, death counts, results, and adjustment for confounding. Hazards ratios, RRs, and standardized mortality rate ratios were treated as equivalent measures of relative mortality risk. Where available, age-adjusted and sex-stratified risks were extracted with preference. The data extracted in the previous review including studies up to February 2013 were re-evaluated for inclusion before merging the original database with the data resulting from the new searches.

### 2.4. Quality Assessment

In line with the original systematic review [18], a quality assessment was performed using the following criteria [25]: representativeness of the sample; measurement and definition of the independent and dependent variables; linkage of survey data; age-adjustment (for details of each criterion see Appendix A). No aggregate score was applied since these quality aspects differ in regard to their importance.

### 2.5. Statistical Analysis

Sex-stratified random-effects DerSimonian-Laird meta-analyses [26] were performed to summarize the RR of dying from an alcohol-attributable cause of death for unemployed individuals compared with employed individuals. Between-study heterogeneity was quantified using the I^2^ statistic [27] and Cochran’s Q [28]. I^2^ was interpreted based on pre-defined guidelines [29]. Potential publication bias was examined using Egger’s regression-based test [30]. We carried out exploratory sensitivity analyses to investigate the impact of overall study quality (all criteria fulfilled versus at least one criterion not fulfilled; Appendix A); and the impact of including causes of death that are less than 100% alcohol-attributable as part of the outcome. Sex-stratified DerSimonian-Laird random-effects meta regression models were used for sensitivity analyses [26]. Analyses were carried out in Stata 15 [31].

## 3. Results

The PRISMA flow chart for study inclusion is shown in Figure 1. A total of ten studies were included in the systematic review, six of which were identified in the review performed in 2013, and four of which were newly identified.

An overview of all studies included in the present meta-analysis is shown in Table 1. In total, the included studies reported findings based on about 14.4 million women and 19.0 million men, among whom there were about 3147 and 17,815 alcohol-attributable deaths, respectively. The studies included data from seven countries, all of which are European or North American high-income countries. With three studies reporting findings from Finland, two from Sweden, and one each from Canada, Lithuania, Poland (Gdansk), the UK (Northern Ireland), and Spain. The studies reported on data spanning over 40 years from 1970 (earliest baseline) up to 2013 (latest follow-up). The included data were obtained from census-linkage (*n* = 8) or longitudinal studies (*n* = 2). Studies differed in regard to their definition of employment status; most studies reported on employed and unemployed individuals (*n* = 6), while others compared professionals and unemployed (*n* = 2), employed, short- and long-time unemployed, or the general population and unemployed individuals. The causes of death included in each of the studies are listed in Appendix A.

### 3.1. Relative Risks for Employment Status

The RRs of dying from an alcohol-attributable cause of death for men and women who were unemployed compared with their employed counterparts were 3.68 (95% CI 2.04–6.66) and 4.93 (95% CI 3.45–7.05), respectively (Figure 2).

Herttua et al. 2008 [20] used short- and long-term unemployment as two levels of unemployment, both of which were included in the meta-analysis. Among both women and men, the higher point estimate refers to long-term unemployment. All studies used “employed” as the reference category, with the exception of Agren and Romelsjö, 1992 [19] and Connolly et al., 2010 [32] who used “professionals” (the highest level of occupation) as the reference group. Sensitivity analysis did not indicate meaningful differences in the resulting point estimate conditional on overall study quality or the inclusion of causes of death that are less than 100% alcohol-attributable (results available upon request).

### 3.2. Heterogeneity and Bias Control

Considerable heterogeneity was detected in the sex-stratified meta-analyses, with an I^2^ > 75% and *p* < 0.01. According to Egger’s weighted regression test, there was no evidence for the presence of publication bias.

## 4. Discussion

The present study reports the most comprehensive pooled risk estimates for alcohol-attributable mortality among unemployed relative to employed individuals using the current evidence, which is largely from Western high-income countries. Our findings indicate that overall, unemployed women have a 3.7-fold higher risk of dying from an alcohol-attributable cause of death compared with employed women. For men, the RR is even higher at 4.9. In contrast to the previous review, which included only two risk estimates [18], the RRs found in the present study (based on ten risk estimates for each of the sexes) are noticeably lower.

According to the present results, unemployed men were found to have a slightly higher alcohol-attributable mortality risk than unemployed women, compared with their employed counterparts. SES is interrelated with gender roles, behaviors, and social expectations [40]. Hence, gender may alter employment–health relationships, with men being more affected by unemployment than women [18,36,41,42]. Since women are traditionally less attached to the labor force [43] and have greater opportunities to switch between rewarding social roles (e.g., care giving) and being employed [44], the consequences of unemployment might affect their health less than it does for men. However, women’s labor force participation has increased substantially during the last few decades.

In many European countries, men drink more frequently and in higher quantities than women [45], possibly explaining the gender differences in our risk estimates. Further, it was found that there are larger differences in the prevalence of risky drinking patterns between men of high and low SES compared with women of high and low SES [46]. Particularly, the frequency of drinking-related health problems was significantly associated with unemployment among men [47]. Contrastingly, unemployment was found to be more strongly related to women’s alcohol consumption [48] and unemployment was associated with increased death rates due to alcohol abuse in women only [44].

Regardless of gender identity, a higher proportion of people in higher SES groups are found to be drinkers consuming smaller amounts of alcohol more frequently, whereas more people in lower SES groups are abstainers but those who do drink do so more often in problematic ways [46,49,50,51,52]. Conclusively, low SES groups tend to drink on fewer occasions but in higher quantities compared with high SES groups [53,54], with the unemployed in particular being likely to consume alcohol at risky levels [55]. While findings on the relationship between SES and drinking patterns are mixed [56,57,58], negative health-related consequences of alcohol use are consistently more prevalent among individuals at the lower end of the socioeconomic spectrum. Thus, both differential exposure and vulnerability may play a role when it comes to heightened mortality risks [59].

In the present study, we demonstrated a link between employment status and alcohol-attributable mortality. However, the underlying mechanisms of this association are not well understood. It has not yet been conclusively determined whether the health state affects a person’s employment status (health selection) or whether a person’s employment status determines health (social causation) [60]. For instance, high SES can serve as a buffer for more frequent and rapid cognitive decline [61] and even prevent disability [62]. On the other hand, there is a reverse causality with individuals with disabilities or serious illness facing unemployment [60]. This is in line with the observation that working individuals demonstrate better health [63], also referred to as the healthy worker effect. It may be even more likely that there is no direct pathway but that other individual-level risk factors are influencing SES and health (indirect selection) [60]). For instance, biological determinants, health literacy and health care, environmental exposure, behavior, and lifestyle [64,65] may moderate the SES-health-relationship. Since we included cohort– and case–control studies as well as cross-sectional studies, with the latter not providing any cause–effect information, this requires further investigation.

### Strengths and Limitations

The presented review is the most comprehensive review of the current literature on the association between employment status and alcohol-attributable mortality risk, including data from ten individual studies. Moreover, we estimated sex-stratified risks, allowing for a more detailed understanding of the association between unemployment and mortality in men and women. However, some limitations have to be noted. First, we included studies that used different definitions of unemployment. While the majority of studies compared risks related to unemployment and employment, the risk estimates of two studies were based on an unemployed–professionals comparison [19,32], and one study compared unemployed individuals with the general population [39]. These differing definitions of the reference group may have influenced the estimated risks. Second, the included studies report on Western high-income countries. The results of the present review are therefore only generalizable to similarly structured countries in the Western world. Since three articles [20,34,35] included Finnish estimates and two studies [19,38] reported on Swedish data, the northern European region was slightly overrepresented. Despite this relative homogeneity of the included countries, there are several contextual modifying factors at both the individual and population level that likely modify the relationship between unemployment and alcohol-attributable mortality. These include, for example, age and marital status at the individual level and welfare policies and the broader economic context at the population level. While this meta-analysis provides a high-level estimate of the risk relationship, important modifying factors need to be investigated in future, country-specific research. Lastly, the operationalization of alcohol-attributable mortality varied between reviewed articles. Whereas most studies included only 100% alcohol-attributable deaths (see Appendix A), few studies [34,35,39] additionally included deaths less than 100% alcohol-attributable (see Appendix A). However, sensitivity analyses did not indicate meaningful differences in the resulting risk estimate conditional on the operationalization of alcohol attributable causes of death.

## 5. Conclusions 

In the present study, we synthesized all available data on the association between employment status and alcohol-attributable mortality risks. We found that unemployed individuals have a high risk of dying from an alcohol-attributable cause. This has important implications for the social support and welfare system, indicating that preventing unemployment may have considerable public health effects. Initiatives such as the Health in All Policies Approach by the World Health Organization [66] serve as a good starting point to tackle this challenge. Additionally, alcohol screening procedures could be directly applied in counselling centers or at primary health care services [55] to identify individuals at risk extensively and as early as possible to refer them to prevention programs or brief interventions. Alcohol control policies such as the SAFER project or minimum-unit pricing [67,68,69,70] and efforts to reduce global unemployment [71,72,73] further pose well-suited initiatives.

## Figures and Tables

**Figure 1 ijerph-19-07354-f001:**
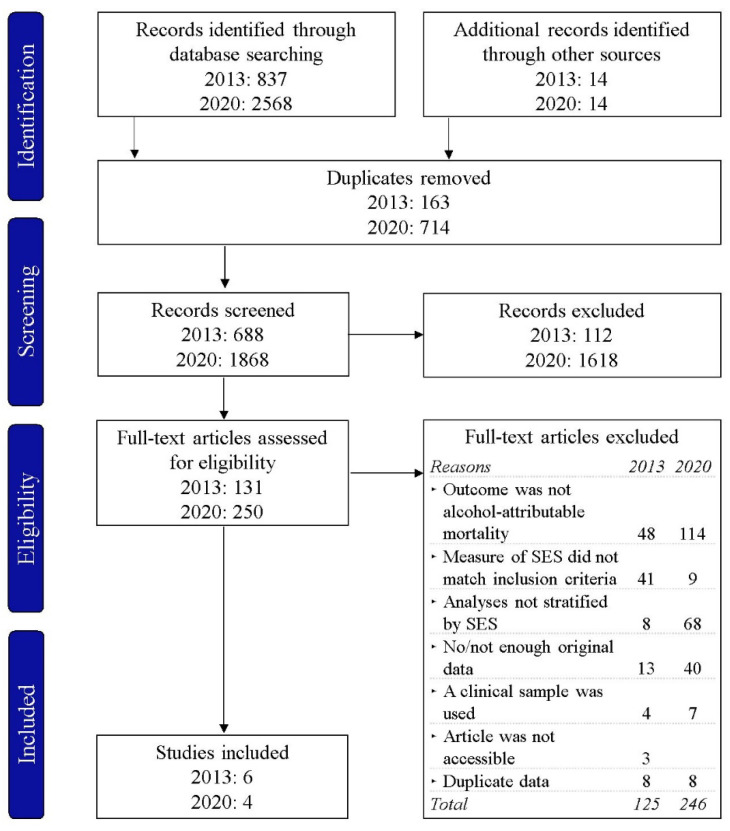
The PRISMA flow chart of the study selection for the search conducted in 2013 (including studies published up to February 2013) and 2020 (including studies published between January 2013 and August 2020). SES, socioeconomic status.

**Figure 2 ijerph-19-07354-f002:**
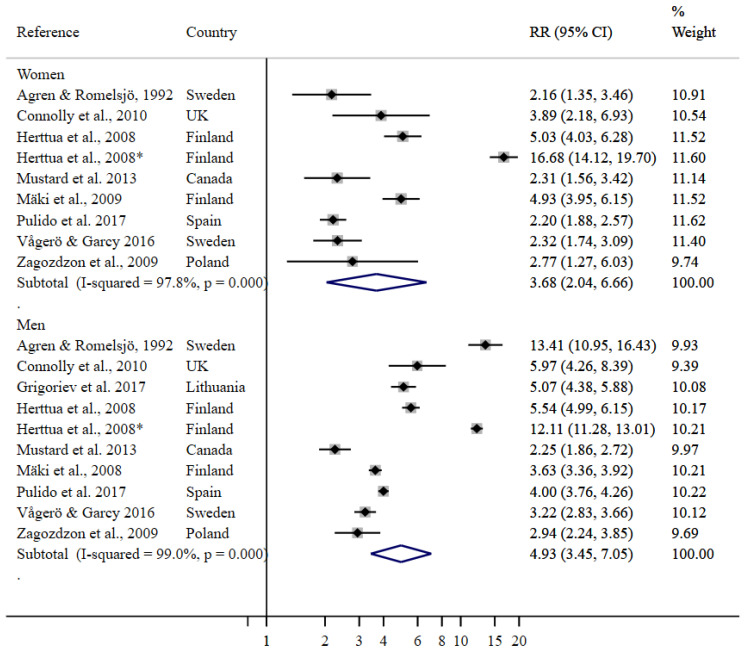
Random-effects meta-regression for employment status. RR, relative risk; CI, confidence interval; UK, United Kingdom. * This estimate is referring to long-term unemployment, whereas the other estimate by Herttua et al. 2008 [20] is referring to short-term unemployment. For women refer to [19,20,32,35,36,37,38,39], for men refer to [19,20,32,33,34,36,37,38,39].

**Table 1 ijerph-19-07354-t001:** The characteristics of all studies included in the sex-stratified random-effect meta-analyses.

Reference	Country, Region/City	Study Years	Study Design	Age Range (Years)	Sample Size by Sex	Number of Deaths by Sex	Employment Status
Agren & Romelsjö, 1992 [19]	Sweden	1970–1975	Census-linkage	25–46	2,008,000 (W), 2,044,000 (M)	405 (W), 2237 (M)	professional, unemployed
Connolly et al. 2010 [32]	UK, Northern Ireland	2001–2006	Longitudinal	25–74	369,245 (W), 351,382 (M)	201 (W), 377 (M)	professional, unemployed
Grigoriev et al. 2017 [33]	Lithuania	2011–2013	Census-linkage	30–64	1,246,000 (M)	1424 (M)	employed, unemployed, inactive/disabled, other inactive
Herttua et al. 2008 [20]	Finland	2000–2003	Longitudinal	30–59	2,018,000 (W), 1,891,000 (M)	555 (W), 2749 (M)	employed, short unemployment, long unemployment
Mäki et al. 2008 [34]	Finland	1990–2001	Census-linkage	25–64	1,051,626 (M)	2703 (M)	employed, unemployed
Mäki et al. 2009 [35]	Finland	1990–2001	Census-linkage	25–64	1,109,497 (W)	563 (W)	employed, unemployed
Mustard et al. 2013 [36]	Canada	1991–2001	Census-linkage	30–69	711,600 (W), 888,000 (M)	207 (W), 926 (M)	employed, unemployed
Pulido et al. 2017 [37]	Spain	2001–2011	Census-linkage	25–64	6,374,624 (W), 9,601,876 (M)	602 (W), 5239 (M)	employed, unemployed
Vågerö & Garcy, 2016 [38]	Sweden	1992–2002	Census-linkage	25–59	1,645,002 (W), 1,747,167 (M)	314 (W), 960 (M)	employed, unemployed
Zagozdzon et al. 2009 [39]	Poland, Gdansk	1999–2004	Census-linkage	20–59 (W), 20–64 (M)	182,387 (W), 185,461 (M)	300 (W), 1200 (M)	general population, unemployed

M, men; W, women.

## Data Availability

The data used in this study are available herein.

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
