# Peer review of "Employment Status and Alcohol-Attributable Mortality Risk—A Systematic Review and Meta-Analysis"

_ijerph, 2022, doi:10.3390/ijerph19127354_

Round 1

Reviewer 1 Report

This is a well-written systematic review on employment status and alcohol-attributable mortality risk. The data is well-presented but I have some comments on the need for a separate review and some of the methods and findings reported in the manuscript.

  1. The authors mention that the current study presents a subset of data from a larger systematic review and meta-analysis. Could the authors explain in the manuscript why a second analysis was conducted and how this is different to the first review and meta-analysis?
  2. In the methods, the authors explain that papers were searched from February 2013 until the last week of August 2020 but in the PRISMA flow chart, it is split by 2013 and 2020. This makes it difficult to interpret as it seems that papers were categorised as being from 2013 or 2020 rather than throughout this period (unless I have misinterpreted this). Please could the search process and flow chart be made more clear.
  3. The relative risks among women reported by Herttua et al., 2008 are much higher compared to other studies included. While a similar observation is noted for Agren & Romelsjo, 1992 and Herttua et al., 2008 among men. Please could the authors provide a comment on why this might have been the case in the results or discussion section. 

Author Response

Thank you for your thorough and helpful review of our manuscript. We appreciate the opportunity to address the reviewers’ suggestions, which we believe have strengthened the manuscript. Below we have provided a point-by-point response to each comment. A clean and a tracked changes version of the manuscript has been uploaded and submitted for consideration. The page numbers referred to below pertain to the clean version of the manuscript.

Reviewer 1

This is a well-written systematic review on employment status and alcohol-attributable mortality risk. The data is well-presented but I have some comments on the need for a separate review and some of the methods and findings reported in the manuscript.

RESPONSE: Thank you for your positive review. The individual comments are addressed in detail below.

  1. The authors mention that the current study presents a subset of data from a larger systematic review and meta-analysis. Could the authors explain in the manuscript why a second analysis was conducted and how this is different to the first review and meta-analysis?

RESPONSE: Thank you for pointing this out. We have now added a sentence in the manuscript. The revised text reads as follows on page 2: “In this first review, we performed dose-response meta-analyses for all indicators of SES except employment status, as it is most often dichotomous (e.g., unemployed vs. employed) it does not lend itself well to a dose-response investigation.”

  1. In the methods, the authors explain that papers were searched from February 2013 until the last week of August 2020 but in the PRISMA flow chart, it is split by 2013 and 2020. This makes it difficult to interpret as it seems that papers were categorised as being from 2013 or 2020 rather than throughout this period (unless I have misinterpreted this). Please could the search process and flow chart be made more clear.

RESPONSE: We agree that this is an important consideration. We have addressed this potential source of confusion by revising the Figure caption as follows: “Figure 1. PRISMA flow chart of study selection for the search conducted in 2013 (including studies published up to February 2013) and 2020 (including studies published between January 2013 and August 2020).”

  1. The relative risks among women reported by Herttua et al., 2008 are much higher compared to other studies included. While a similar observation is noted for Agren & Romelsjo, 1992 and Herttua et al., 2008 among men. Please could the authors provide a comment on why this might have been the case in the results or discussion section. 

RESPONSE: As stated in the manuscript we have included two point estimates from Herttua et al. 2008. We have now added an asterisk in Figure 2 with the following explanation: “* This estimate is referring to long-term unemployment, whereas the other estimate by Herttua et al. 2008 is referring to short-term unemployment.”

One reason underlying the heterogeneity observed for Agren and Romelsjö may be the comparison to professionals rather than all employed individuals, as noted in the manuscript, e.g., in the Discussion section:

“While the majority of studies compared risks related to unemployment and employment, the risk estimates of two studies were based on an unemployed-professionals-comparison [19, 32] and one study compared unemployed individuals to the general population [39].” (line 245)

Reviewer 2 Report

The study authors review existing research on the relationship between unemployment and alcohol-attributable mortality risk. The secondary analysis, in effect, looks at ten existing studies. Although we have information about the countries to which the studies apply at the beginning of the text, it is only in the conclusion that we have a declaration of the authors' self-awareness that the results apply only to a fragment of the Western world. It seems that the geographical scope of the research should have been clearly defined, especially since the authors sometimes give the impression that they are drawing globally valid conclusions. Out of ten studies, three concern Finland and two concern Sweden. The UK study refers only to Northern Ireland and the Polish study refers - according to the authors - to one city, which is not representative for the country. The title of the Polish study suggests, however, that it is a nationwide study, but concerns the factor of political transformation which is absent in the reviewed text.
The topic of COVID is mentioned in the text, although it is not clear why, as it is not reflected in further analysis.
The authors are aware of the limitations of the presented results, which only concern the relationship between unemployment, alcohol-attributable mortality risk and gender.
What is missing, however, is a question about the national variation in these correlations due to tradition, culture or climate (lack of sunlight). In particular, cultural background or marital status are not included. Also, no differentiation was made according to the age of the respondents. Some of the ten empirical studies cited refer to the 30-64 age group, while others to the 20-59 age group. It seems reasonable to assume that the relationship between unemployment and the risk of death from alcohol varies across the different age groups, if only because of the rather low probability of deep alcohol dependence among 20-year-olds (It may be that this last point is not true, but this would need to be demonstrated by empirical studies).
It also did not take into account what type of death is involved in the risk studied. Is it suicide, alcohol poisoning, alcohol-related accident, etc.? It is only noted that some studies refer only to 100% alcohol-attributable deaths and others to less than 100% alcohol-attributable deaths. This refers us to Table S2 and S3, while in the text we have only Table 1.

The authors are aware of the other limitations of their study.

Author Response

Thank you for your thorough and helpful review of our manuscript. We appreciate the opportunity to address the reviewers’ suggestions, which we believe have strengthened the manuscript. Below we have provided a point-by-point response to each comment. A clean and a tracked changes version of the manuscript has been uploaded and submitted for consideration. The page numbers referred to below pertain to the clean version of the manuscript.

Reviewer 2

The study authors review existing research on the relationship between unemployment and alcohol-attributable mortality risk. The secondary analysis, in effect, looks at ten existing studies. Although we have information about the countries to which the studies apply at the beginning of the text, it is only in the conclusion that we have a declaration of the authors' self-awareness that the results apply only to a fragment of the Western world. It seems that the geographical scope of the research should have been clearly defined, especially since the authors sometimes give the impression that they are drawing globally valid conclusions. Out of ten studies, three concern Finland and two concern Sweden. The UK study refers only to Northern Ireland and the Polish study refers - according to the authors - to one city, which is not representative for the country. The title of the Polish study suggests, however, that it is a nationwide study, but concerns the factor of political transformation which is absent in the reviewed text.

RESPONSE: Thank you for this comment. We agree that it is important to not draw global conclusions from the geographical subregion represented by the ten included studies. As the study protocol and search strategy were not focussed on identifying studies from one region only (no language restrictions were applied although the search was performed in English and the vast majority of research papers available through Pubmed is in English), we decided not to specify a geographical scope of the study from the outset. However, we have now added a further sentence to highlight the geographical focus of the studies that we identified in the results section before listing the countries: “The studies included data from seven countries, all of which are European or North American high-income countries.” (line 147) To maintain the separation between description and interpretation of the results, we did not add any further comments.

Furthermore, we address the limited geographical scope in the first sentence of the discussion section: “The present study reports the most comprehensive pooled risk estimates for alcohol-attributable mortality among unemployed relative to employed individuals using the current evidence, which is largely from Western high-income countries”. Furthermore, we revised the respective paragraph in the limitations section as follows: “Second, the included studies report on Western high-income countries. The results of the present review are therefore only generalizable to similarly structured countries of the Western world. Since three articles [20, 34, 35] included Finnish estimates and two studies [19, 38] reported on Swedish data, the Northern European region was slightly overrepresented.”

The topic of COVID is mentioned in the text, although it is not clear why, as it is not reflected in further analysis.

RESPONSE: We have discussed this point internally, acknowledging the reviewer’s argument that COVID-19 is not part of the study. However, we still felt that it was worthwhile providing a reference to current developments and exacerbated challenges of the labour force by COVID to further underline the relevance of our research. Given the rise in the unemployment rate due to COVID-19, the risks presented become more pertinent, thus possibly prompting additional studies on the risk of alcohol-attributable causes of death among the unemployed.

The authors are aware of the limitations of the presented results, which only concern the relationship between unemployment, alcohol-attributable mortality risk and gender.
What is missing, however, is a question about the national variation in these correlations due to tradition, culture or climate (lack of sunlight). In particular, cultural background or marital status are not included. Also, no differentiation was made according to the age of the respondents.

RESPONSE: We agree that this is a potential limitation of the study. Unfortunately, four of the ten reviewed studies did not provide data on marital status and five of the studies did not examine death rates depending on cultural background. Furthermore, the study base is yet too sparse to systematically investigate reasons underlying the observed heterogeneity in the point estimates. We have added the following sentence to the limitations section to acknowledge the inability to account for potential covariates: “Despite this relative homogeneity of the included countries, there are several contextual modifying factors at both the individual- and population-level that likely modify the relationship between unemployment and alcohol-attributable mortality. These include, for example, age and marital status at the individual-level and welfare policies and the broader economic context at the population-level. While this meta-analysis provides a high-level estimate of the risk relationship, important modifying factors need to be investigated in future, country-specific research.” (line 250)

Some of the ten empirical studies cited refer to the 30-64 age group, while others to the 20-59 age group. It seems reasonable to assume that the relationship between unemployment and the risk of death from alcohol varies across the different age groups, if only because of the rather low probability of deep alcohol dependence among 20-year-olds (It may be that this last point is not true, but this would need to be demonstrated by empirical studies).

RESPONSE: Thank you for this thoughtful comment. To adjust for difference in age distribution, we used age-adjusted estimates to calculate the RRs whenever available. Reviewing Conolly et al., 2010, regrettably, only sex-stratified estimates could be used. Furthermore, we have added a short paragraph to the limitations section, cited above.

It also did not take into account what type of death is involved in the risk studied. Is it suicide, alcohol poisoning, alcohol-related accident, etc.? It is only noted that some studies refer only to 100% alcohol-attributable deaths and others to less than 100% alcohol-attributable deaths. This refers us to Table S2 and S3, while in the text we have only Table 1.

RESPONSE: We agree that a potential limitation is that varying type of deaths were included in the studied risks. However, the studies indeed had a very high overlap. It should be noted that all studies had to include predominantly 100% alcohol-attributable causes of death but studies that additionally included some causes that are not a 100% attributable were also eligible. We have now adapted the respective sentence in the methods section as follows: “Alcohol-attributable causes of death were defined as all underlying causes of death that are fully attributable to alcohol use [23]. Studies that included causes with an alcohol-attributable fraction (AAF) of at least 10% globally [24] (see Table S2, and Table S3 for ICD-codes) in addition to 100% attributable causes were also eligible.” (line 97)

We have also added a table to the appendix (Table S7) showing the causes of death along with ICD codes (where available) for each of the included studies. Finally, it should be noted that the sensitivity analysis did not indicate any meaningful differences in the resulting point estimate conditional on inclusion of causes of death that are less than 100% alcohol-attributable.

The authors are aware of the other limitations of their study.

Reviewer 3 Report

In my opinion, although the study has good intentions, it lacks a suitable methodological study to be able to meet the proposed objectives, obtaining results that, in my opinion, do not achieve the proposed objective in a suitable way.

The article deals with the relationship between employment status and the risk of alcohol-attributable mortality through a meta-analysis of the studies previously carried out on this subject.

The studies on this subject are diverse and it is difficult to combine the specific conditions to obtain suitable results and, in my opinion, the way the meta-analysis has been carried out does not manage to offer suitable conclusions.

Author Response

Thank you for your thorough and helpful review of our manuscript. We appreciate the opportunity to address the reviewers’ suggestions, which we believe have strengthened the manuscript. Below we have provided a point-by-point response to each comment. A clean and a tracked changes version of the manuscript has been uploaded and submitted for consideration. The page numbers referred to below pertain to the clean version of the manuscript.

Reviewer 3

In my opinion, although the study has good intentions, it lacks a suitable methodological study to be able to meet the proposed objectives, obtaining results that, in my opinion, do not achieve the proposed objective in a suitable way.

RESPONSE: We regret that the reviewer is of the opinion that the study does not achieve its objective. We hope that our revisions have clarified both suitability of our approach, as well as the objective. As stated by Reviewer 2, the review cannot provide a definite estimate for the global context nor can it account for all potential covariables and effect modifiers. However, we are convinced that it can provide a rough overall estimate on the risk relationship between employment status and alcohol-attributable mortality. We have modified the objective as follows to emphasize that we are aiming to summarize the available evidence: “Therefore, the present work aims to provide updated, sex-stratified estimates on the relative alcohol-attributable mortality risk depending on employment status based on the current evidence.”

The article deals with the relationship between employment status and the risk of alcohol-attributable mortality through a meta-analysis of the studies previously carried out on this subject.

The studies on this subject are diverse and it is difficult to combine the specific conditions to obtain suitable results and, in my opinion, the way the meta-analysis has been carried out does not manage to offer suitable conclusions.

RESPONSE: While we agree that there is heterogeneity in the country contexts that potentially modify the risk relationship, we are convinced that the included studies are sufficiently homogenous to allow for a statistical summary. All included studies are based on the adult, general population, the operationalization of alcohol-attributable mortality is similar across studies (as now evidenced by the newly added Table S7), and the exposure of being unemployed is measured at the individual level with good comparability. Beyond that, we have now added a new paragraph to the discussion section to acknowledge the limitations of the current approach to account for potential sources of heterogeneity: “Despite this relative homogeneity of the included countries, there are several contextual modifying factors at both the individual- and population-level that likely modify the relationship between unemployment and alcohol-attributable mortality. These include, for example, age and marital status at the individual-level and welfare policies and the broader economic context at the population-level. While this meta-analysis provides a high-level estimate of the risk relationship, important modifying factors need to be investigated in future, country-specific research.“ (line 250)

Round 2

Reviewer 2 Report

The text of the article has been substantially revised. It may be accepted for publication in its present form.

Reviewer 3 Report

 here is the review of the article
